# Polymicrobial Infections in the Immunocompromised Host: The COVID-19 Realm and Beyond

**DOI:** 10.3390/medsci10040060

**Published:** 2022-10-20

**Authors:** Eibhlin Higgins, Aanchal Gupta, Nathan W. Cummins

**Affiliations:** Division of Public Health, Infectious Diseases and Occupational Medicine, Mayo Clinic, Rochester, MN 55902, USA

**Keywords:** transplant infectious diseases, immunocompromised host, COVID-19, polymicrobial infection

## Abstract

Immunosuppression changes both susceptibility to and presentation of infection. Infection with one pathogen can also alter host response to a different, unrelated pathogen. These interactions have been seen across multiple infection domains where bacteria, viruses or fungi act synergistically with a deleterious impact on the host. This phenomenon has been well described with bacterial and fungal infections complicating influenza and is of particular interest in the context of the COVID-19 pandemic. Modulation of the immune system is a crucial part of successful solid organ and hematopoietic stem cell transplantation. Herein, we present three cases of polymicrobial infection in transplant recipients. These case examples highlight complex host–pathogen interactions and the resultant clinical syndromes.

## 1. Introduction

The interaction of host and pathogen is fundamental to the pathogenesis of infectious diseases. This interaction is markedly different in the setting of immunosuppression resulting in altered susceptibility, presentation and severity of infection (Figure 1). The evolution of immunosuppressive therapy has been paramount to the success of both solid organ and stem cell transplantation but understanding how these manipulate host infection risk is crucial to post-transplant care. Infection with one pathogen can also alter the host response to other pathogens. This may be related to modulation in the immune response or direct effect of the pathogen on other protective barriers such as skin, mucosa or respiratory epithelium. Coinfection with two or more unrelated pathogens has been described in a multitude of clinical scenarios. Bacterial pneumonia complicating influenza is well described although the reported prevalence varies widely [1] and is associated with poorer clinical outcomes [2]. Fungal coinfection can also occur with respiratory viruses, and this phenomenon is seen both with influenza [3] and more recently with SARS-CoV-2 [4]. In immunocompromised individuals, particularly in the transplant population, Cytomegalovirus (CMV) is an important cause of morbidity and mortality. CMV augments the immunosuppressive state and has been reported to increase susceptibility to infection with other herpesviruses as well as fungal and bacterial pathogens [5]. In this case series, we review three cases demonstrating pathogen coinfection in immunocompromised hosts, a population where Occam’s razor may not always apply.

## 2. Case Series

### 2.1. Case One

A 60-year-old female presented to the emergency department 12 weeks post deceased donor liver transplantation undertaken for decompensated cirrhosis with hepatorenal syndrome. Two weeks prior to presentation, she noted the onset of dry cough, chills and myalgia. After initial improvement in symptoms with supportive care, she subsequently developed progressive shortness of breath, worsening cough with sputum production, fevers and diarrhea.

Prior to her transplantation, serologies revealed a positive IgG for Epstein–Barr Virus (EBV) and CMV. She received basiliximab as induction immunosuppression followed by maintenance therapy with mycophenolate mofetil, prednisone and tacrolimus. Her post-transplant course was complicated by hematoma requiring surgical evacuation, acute kidney injury requiring hemodialysis and an episode of acute cellular rejection treated with IV methylprednisolone. Previous medical history also included hypertension and chronic kidney disease.

At the time of presentation, she was hemodynamically stable and afebrile. Her physical exam was notable for left sided rhonchi and a well-healed abdominal incision. Her white blood cell count demonstrated leukopenia with a total count of 1.1 × 10^9^/L (normal range 3.4–9.6 × 10^9^/L), neutrophils of 0.33 × 10^9^/L (normal range 1.56–6.45 × 10^9^/L) and lymphocytes of 0.5 × 10^9^/L (normal range 0.95–3.07 × 10^9^/L). She had an acute chronic kidney injury with creatinine of 4.39 mg/dL (range 0.59–1.04 mg/dL) and a normal liver profile. Computed tomography (CT) of the chest showed multiple peribronchial nodules and scattered ground-glass opacities. A nasopharyngeal swab was sent and tested with a multiplex polymerase chain reaction (PCR) respiratory pathogen panel. This was positive for rhinovirus. Quantitative CMV PCR on serum was negative. Sputum cultures were positive for methicillin susceptible *Staphylococcus aureus* (MSSA). Admission blood cultures returned positive for MSSA after 63 h and repeat surveillance blood cultures were negative. A stool sample was later sent due to persistent diarrhea and tested positive for *Clostridium difficile* toxin. Urine antigen testing was positive for *Histoplasma* antigen, but it was below the limit of quantification, and subsequent serum antibody and antigen testing for *Histoplasma* was negative. She received a 14-day course of IV cefazolin for MSSA pneumonia with associated bacteremia and was treated with oral vancomycin for *Clostridium difficile* infection (CDI).

Whilst human rhinoviruses are an established cause of the ‘common cold’ in immunocompetent individuals, they are more commonly associated with lower respiratory infection than previously thought [6]. In a retrospective study of symptomatic hematopoietic stem cell transplant recipients, isolation of rhinovirus from either the upper or lower respiratory tract was associated with a 90-day mortality of 6% and 41%, respectively [7]. There is limited data on the frequency of bacterial coinfection with rhinovirus in transplant recipients, but in a retrospective study in the pediatric population, detection of rhinovirus on nasopharyngeal swabs increased the rate of colonization with both *Streptococcus pneumoniae* and *Staphylococcus aureus* [8]. Experimental models of human rhinovirus infection have also demonstrated changes in upper respiratory tract microbiota during acute infection [9]. Rhinovirus has also been shown to increase internalization of *Staphylococcus aureus* into epithelial cells in vitro; another putative mechanism for coinfection [10]. The time course of our patient’s presentation correlates with symptomatic rhinovirus infection and subsequent development of bacterial pneumonia confirmed by radiologic imaging and isolation of MSSA from sputum sample and blood cultures. Antibiotic exposure is the most important risk factor for CDI, but immunosuppression and healthcare exposure [11] were also important contributing factors in this case. Although urinary *Histoplasma* antigen was detected, it was below the level of quantification, and this was felt to be a false positive in light of both an alternative explanation for the presentation along with a negative serum antibody and antigen. A previous study of patients with weakly positive urinary *Histoplasma* antigen demonstrated that in 48% of cases they were determined to be false positives [12]. Whilst the numbers in this study were small, it does highlight the importance of the clinical correlation of laboratory data.

### 2.2. Case Two

A 61-year-old male presented to the emergency department with a 7-day history of a progressive cough with associated dyspnea, pleuritic chest pain and occasional chills. About 4 weeks prior to current hospitalization, he was diagnosed with COVID-19 pneumonia for which he initially received casirivimab-imdevimab followed by course of remdesivir and dexamethasone due to worsening of symptoms. He had a history of myelofibrosis for which he underwent an HLA matched unrelated donor allogeneic hematopoietic stem cell transplant (HCT) after conditioning with fludarabine and melphalan, approximately a year prior to current hospitalization. His CMV IgG was positive pre-transplant, and donor CMV serology was negative.

His post-transplant course was complicated by colonic GVHD for which he received tacrolimus and budesonide as well as recurrent CMV viremia treated with valganciclovir. He was on antibacterial/antifungal prophylaxis with doxycycline, posaconazole and trimethoprim-sulfamethoxazole. Other medical history included hypogammaglobulinemia and splenectomy in the setting of MDS, type 2 diabetes mellitus, congestive heart failure and hypertension.

On initial evaluation, he was afebrile, in mild respiratory distress but was not hypoxemic. Labs revealed a normal white blood cell count, chronic anemia with hemoglobin of 9.9 g/dL (normal range 11.6–15 g/dL, creatinine of 1.34 mg/dL (baseline of 1.08, normal range 0.59–1.04 mg/dL), d-dimer of 2410 ng/mL (normal range ≤ 500 ng/mL fibrinogen equivalent units), normal ferritin and C-reactive protein. CT angiography of the lung showed improvement in prior bilateral ground-glass and consolidative opacities. A nasopharyngeal swab was sent, and multiplex PCR pathogen panel was positive for coronavirus OC43 strain, parainfluenza 4 virus and SARS-CoV-2. He improved with supportive care and was subsequently discharged.

Human parainfluenza virus (HPIV) is an RNA virus from Paramyxoviridae family. HPIV infection in allogeneic HCT recipients is associated with lower respiratory tract infection (LRTI) in 30% of cases with an associated mortality of 27% [13]. It has also been shown to increase the risk of late onset airflow obstruction in HCT recipients and chronic rejection in lung transplant recipients [14]. Human coronaviruses (HCoV) belong to Coronaviridae family, with four different species: HCoV-229E, -NL63, -OC43 and -HKU1. The mainstay of therapy is supportive care, although there has been reports of use of IVIG or ribavirin for parainfluenza virus [15]. The phenomenon of co-infection or superinfection with other viruses, bacteria or fungi is being increasingly reported during the COVID-19 pandemic, though the rate remains relatively low with one study demonstrating 8% rate of bacterial and fungal community-acquired co-infections [16], and another study citing a rate of 2.1% for bacterial and 0.6% for viral co-infections [17] during the early stages of the pandemic. A metanalysis demonstrated a pooled prevalence of viral co-infections of 5% with influenza and rhinovirus being the most common, followed by adenovirus and human coronaviruses [18]. Studies have demonstrated increased severity of COVID-19, more frequent ICU admissions and higher risk of death in co-infected patients [17,18,19,20,21]. Incidence of non-SARS-CoV-2 respiratory viruses was significantly impacted by social distancing measures at the start of the pandemic. With the phased reduction in such measures leading to expected increased circulation of such viruses, our understanding of co-infection will continue to broaden.

### 2.3. Case Three

A 32-year-old female presented to the emergency department with a 5-day history of fevers, chills, myalgias and worsening shortness of breath. She underwent deceased donor liver transplant for fulminant liver failure in setting of autoimmune hepatitis four years previously. Serologies demonstrated positive CMV IgG in the donor and negative CMV IgG in the recipient. Her post-transplant course was complicated by low-level delayed primary CMV viremia and multiple episodes of acute cellular rejection. She was on maintenance immunosuppression with tacrolimus, azathioprine and low-dose prednisone. She also had a history of type 2 diabetes mellitus requiring insulin therapy. She had received three doses of mRNA vaccine against COVID-19 and tixagevimab-cilgavimab as pre-exposure prophylaxis against COVID-19.

She was hypoxemic at presentation with an oxygen saturation of 90% requiring 2 L of supplemental oxygen. Labs demonstrated leukopenia of 0.9 × 10^9^/L (normal range 3.4–9.6 × 10^9^/L), neutrophils of 0.56 × 10^9^/L (normal range 1.56–6.45 × 10^9^/L), lymphocytes of 0.3 × 10^9^/L (normal range 0.95–3.07 × 10^9^/L), platelets of 97 × 10^9^/L (normal range 157–371 × 10^9^/L), acute kidney injury with creatinine of 1.36 mg/dL (normal range 0.59 to 1.04 mg/dL), D-Dimer of (normal range <= 500 ng/mL fibrinogen equivalent units), ferritin of 1056 ng/mL (normal range 11–307 mcg/L), c-reactive protein of 312 mg/L (normal ≤ 8.0 mg/L) and lactate dehydrogenase of 474 U/L (normal range 122–222 U/L). A nasopharyngeal swab was positive for SARS-CoV-2 and Respiratory Syncytial Virus (RSV) on PCR. CT angiography of lungs demonstrated patchy airspace opacities bilaterally with large right lower lobe consolidation so the patient was started on ceftriaxone and azithromycin for possible superimposed community acquired bacterial pneumonia. Two sets of blood cultures grew *Streptococcus pneumoniae* after 11 h. Subsequent surveillance blood cultures were negative. She was also treated with a 5-day course of remdesivir and dexamethasone for concomitant COVID-19 pneumonia. She had improvement in cytopenias, inflammatory markers and was weaned off supplemental oxygen by hospital day 5, but after initial improvement, she was re-admitted about 3 days later with worsening respiratory symptoms and found to have large right parapneumonic effusion with pleural fluid analysis consistent with empyema. Pleural fluid cultures showed no microbial growth. She was re-started on ceftriaxone and underwent image-guided chest tube placement but ultimately required open decortication. She received a 4-week course of ceftriaxone with follow up chest radiograph showing near resolution of right-sided consolidation and effusion.

This complex case demonstrates the clinical consequences of the interaction of an immunocompromised host and three separate pathogens. Her clinical syndrome was consistent with COVID-19 complicated by *Streptococcus pneumoniae* pneumonia, blood stream infection and ultimately empyema. Surveillance data from Public Health England [22] suggests the co-occurrence of invasive pneumococcal disease (IPD) and COVID-19 infection is rare. However, when it occurred in the studied cohort it was associated with a 7.8-fold (95% CI, 3.8–15.8) higher case fatality rate than IPD alone [22]. A similar pathogen synergism with deleterious host effect is seen with influenza and pneumococcal disease. Contributing factors include damage to pulmonary epithelium, increased bacterial adherence and exaggerated innate immune response [23]. The extent to which RSV contributed to this case presentation is unclear; it can be associated with lower respiratory tract infection in transplant recipients, although this is more commonly encountered in lung transplant recipients and in allogeneic HCT recipients [15]. Coinfection with SARS-CoV-2 and influenza A in an animal model has been shown to prolong the primary virus infection period, increase immune cell infiltration in bronchoalveolar lavage fluid and result in more severe lymphopenia in peripheral blood with reduced neutralizing antibody titers and CD4^+^ T cell responses against each virus [24]. Whether similar effects may be seen with SARS-CoV-2 and RSV has not yet been fully elucidated. However, such viral synergism could have contributed to the profound lymphopenia seen in this case. The story of the interplay between SARS-CoV-2 and coinfections continues to evolve. SARS-CoV-2 infection results in changes in the respiratory [25] and gut [26] microbiome. The resulting dysbiosis and immune dysregulation likely impacts both disease severity and risk of coinfection. The reported incidence of bacterial co-infection or superinfection is variable, and there is significant heterogeneity in the diagnostic criteria utilized and the cohort included in the literature. Whilst the overall risk of coinfection with SARS-CoV-2 is relatively low, it appears to be more frequent in solid organ transplant recipients [27]. It is associated with worsening of clinical outcomes [28] and can, as demonstrated in this case, result in significant morbidity.

## 3. Discussion

In this case series, we appraised three cases in immunocompromised hosts where the clinical presentation was attributable to infection with multiple pathogens. The pathogen–host interaction is altered not only by host immunosuppression related to solid organ/stem cell transplant but also by the effect of the primary pathogen on the host. Co-infection with multiple pathogens may be fostered or promoted through several different mechanisms (summarized in Table 1) and is primarily related to changes in host barrier function, alteration in immune response and synergistic virulence factors.

Pathogen co-infection or superinfection is of particular interest in the COVID-19 era where immunomodulation with corticosteroids, IL-6 receptor antagonists and small molecule inhibitors form part of the standard treatment algorithm [33]. Current guidelines are largely founded on clinical trials where the majority of patients were not immunosuppressed at baseline. The extent to which further immunomodulation in immunocompromised hosts increases the risk of coinfection is not well established. In this subgroup, treatment decisions draw upon available evidence as well as individualized risk–benefit analyses.

There have been significant improvements in microbiological techniques in recent years with the advent of molecular diagnostics. With the use of PCR and more recently metagenomic sequencing, our capacity for pathogen detection has significantly improved. This comes with a risk of false positive results and the detection of micro-organisms which are not contributing to the clinical syndrome. Thus, whilst immunocompromised hosts may be at higher risk of polymicrobial infection, the clinical context of diagnostic results remains paramount to avoid overtreatment, unnecessary antimicrobial exposure and adverse drug effects.

## 4. Conclusions

In this cases series, the pathogen–host interaction is markedly altered due to immunosuppression either for solid organ or hematopoietic stem cell transplantation. These clinical scenarios demonstrate how polymicrobial infection can impact presenting features, disease severity and therapeutics. The reported prevalence of coinfection or superinfection in the transplant population is variable and likely dependent on the extent of immunosuppression, organ transplanted, timing post transplantation and nature of initial infection. It has been associated in multiple domains with increased morbidity and mortality [28,34]. The resultant morbidity and complex clinical syndrome which may ensue when pathogens act synergistically in immunocompromised hosts is clearly outlined by these three case examples and highlights the need for a broad differential even when one infectious diagnosis has been established.

## Figures and Tables

**Figure 1 medsci-10-00060-f001:**
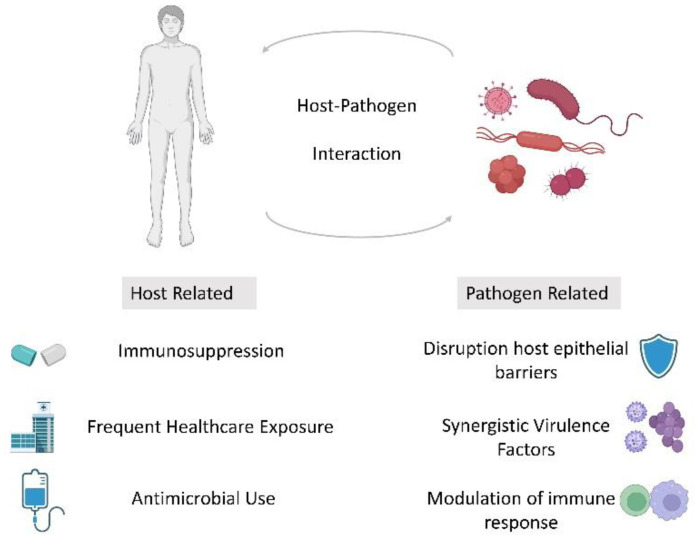
Schematic of factors which may contribute to polymicrobial infection in the immunocompromised host.

**Table 1 medsci-10-00060-t001:** Summary of mechanisms contributing to co-infections.

**Bacterial Coinfections in Respiratory Viral Disease** [25,29]	- Alteration in host microbiome - Epithelial damage enhancing bacterial adhesion - Dysregulation of protective immune response - Synergism of viral and bacterial virulence factors
**Fungal Coinfections in Respiratory Viral Disease** [30,31]	- Cilial dysfunction may disrupt clearance of fungal spores from upper airways - Epithelial dysfunction at alveolar level may allow fungal spores to escape endosomal-lysosomal system and germinate - Inflammation/obstruction of small airways, reduced diffusion capacity with resultant hypoxia - Alterations in innate and adaptive immune response
**Viral Coinfections** [32]	- Impaired cell mediated immunity - Upregulation of proinflammatory cytokines - Modulation of viral replication

## Data Availability

Data sharing is not applicable to this article. No new data were created or analyzed in this study.

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
