# Peer review of "Polymicrobial Infections in the Immunocompromised Host: The COVID-19 Realm and Beyond"

_medsci, 2022, doi:10.3390/medsci10040060_

Round 1

Reviewer 1 Report (New Reviewer)

The case series “Polymicrobial Infections in the Immunocompromised Host: the COVID-19 realm and beyond” by Eibhlin Higgins et al, is an interesting one. The authors have provided a detailed account of the multiple infections the three patients suffer due to their immunocompromised condition. They have also analysed the implications, the diagnostic procedure and treatment methods in the situation of post-transplant complications arising due to polyinfections. In all three cases discussed in the report, immunosuppression had a great impact in patients contracting various viral and microbial infections. Whereas it is absolutely critical to make the correct diagnosis, it is also vital that the treatment method needs to be streamlined so that it can take care of the infections without causing further complications to the patient. Immunosuppression is a great option for organ or hematopoietic stem cell transplant. But, the problems that come up following this procedure need great clinical attention for the benefit of the patients. The authors have discussed these issues very well in the three cases they have presented.

Host-pathogen interaction is a critical determinant of the disease progression. There are multiple ways that the initial interaction could either improve or exacerbate the patient’s conditions. As the authors have mentioned in these cases, immunosuppression led to complications in other organ systems in all three patients. This issue is of greater importance in the prevailing situation of Covid-19 infection. The treatment options with immunosuppression in Covid-19 patients could rapidly aggravate their overall situation and which was the case in the case #3.

The authors have correctly indicated that the improvements in the molecular diagnostic procedures will be very helpful in the management of patients following post-transplant polyinfections.

The report is well written and well analysed.

Author Response

We thank the reviewer for their comments and expert review of our manuscript

Reviewer 2 Report (New Reviewer)

The authors should clearly formulate an hypothesis at the end of the introduction, which they are testing with the three case studies. In the concluding remarks, they should highlight whether the studied cases support the initial hypothesis or prove it wrong.

Although this manuscript is a case report, it is important to clearly state the aims of the study and whether they could be fulfilled.

Is Figure 1 original? Could the authors improve its resolution?

Minor comments:

Please replace "microbiologic techniques" with "microbiological techniques".

Author Response

We thank the reviewer for their expert review and commentary. As a case series we have not written or designed this to test a hypothesis. However, we appreciate the reviewer's comments. The introduction ends with a statement 'we review three cases demonstrating pathogen coinfection in immunocompromised hosts- a population where Occam’s razor may not always apply', In each case we then present how the simplest/single explanation or pathogen did not account for the presentation. We have added a conclusion in which we re-iterate that polymicrobial infection can alter presentation, disease severity and necessary therapeutics. 

The image is an original figure created using Biorender software which we have now acknowledged at the end of the text and we have improved the resolution.

 "microbiologic techniques"  has been replaced with "microbiological techniques"

Round 2

Reviewer 2 Report (New Reviewer)

The authors have satisfactorily answered my comments.

Nevertheless, Figure 1 still has low resolution in my opinion. Maybe the authors can send the image directly for production and not insert it in text. Please beware that it is now inserted twice in the manuscript.

This manuscript is a resubmission of an earlier submission. The following is a list of the peer review reports and author responses from that submission.

Round 1

Reviewer 1 Report

This paper highlights the risk of polymicrobial infection in the immunocompromised hosts. The authors present 3 case reports to illustrate this phenomenon. These case reports are very well described but no experimental data were proposed to explain such interaction between pathogens (bacteria, viruses, and fungi). Following the different cases, the authors give some explanation on the putative mechanisms contributing to co-infections, but it lacks a connection between clinical observation and the pathophysiological mechanisms proposed. This paper could have been improved if more fundamental data could support the microbiological data of each case report. Even if these case reports are well written, this paper bring not enough new data to be published as it stands.

p2 : Cytomegalovirus without uppercase

Reviewer 2 Report

The manuscript "Polymicrobial Infections in the Immunocompromised Host: the COVID-19 realm and beyond" describes 3 cases of patients underwento to solid organ or HSC transplant with multiple contemporary infections during the early or late follow up.

This scenario is well known in the transplant setting. The paper do not add any innovative knowledge to the readers who have expertize in the tranplant field.

Reviewer 3 Report

In this article, Higgins et al., describe three case studies in which individuals with transplantation history present with multiple pathogenic infections. The authors briefly describe the status of the patients in terms of their disease signs and symptoms, prior history of transplantation, treatments etc. In each of these cases, the patients appear positive for at least two distinct pathogens which makes the diagnosis and intervention strategies complex. The authors argue that the immunosuppression induced due to transplantation related therapies as well as infections, may be responsible for the patient to be highly susceptible to coinfections with opportunistic pathogens.

This is a nicely written article which highlights a complicated subject of how immunosuppression together with polymicrobial interaction is relevant but underappreciated in clinical settings. The authors cite appropriate references to make their point and outline a general schematic representing these complex interactions. The extent to which one pathogen contributes to the overall disease is often unclear in clinical settings and hence accurate diagnosis and proper intervention is vital. In that sense, the article highlights key piece of information. This reviewer does not have any major points of critique for this article and agrees with the overall line of thought the authors present.

However, COVID-19 associated changes in gut dysbiosis also seems to contribute to overall health. These changes often reflect in less diverse microbiome composition and can be included as one of the factors in the schematic. In addition, since first two cases are in older adults, age is another important factor that may contribute to immune dysfunction. This will provide a more wholistic view of how polymicrobial interactions with host related changes affect the clinical outcome of disease in transplant patients.